# Synthesis, Characterization of a New Polyacrylic Acid Superabsorbent, Some Heavy Metal Ion Sorption, the Adsorption Isotherms, and Quantum Chemical Investigation

**DOI:** 10.3390/ma13194390

**Published:** 2020-10-01

**Authors:** Sevil Savaskan Yilmaz, Nuri Yildirim, Murat Misir, Yasin Misirlioglu, Emre Celik

**Affiliations:** 1Department of Chemistry, Faculty of Sciences, Karadeniz Technical University, University Avanue, 61080 Trabzon, Turkey; nuri@ktu.edu.tr (N.Y.); celkemre@ktu.edu.tr (E.C.); 2Faculty of Engineering and Architecture, Ahi Evran University, 40100 Kırşehir, Turkey; murat.misir@ahievran.edu.tr (M.M.); nisayzal@gmail.com (Y.M.)

**Keywords:** polyacrylic acid superabsorbent polymer (PAA SAP), sorption capacity, swelling ratio, distribution ratio, Langmuir and Freundlich isotherms, conformational analysis

## Abstract

Poly(acrylic acid/Kryptofix 23-Dimethacrylate) superabsorbent polymer [P (AA/Kry23-DM) SAP] was synthesized by solution polymerization to remove Co, Ni, Cu, Cd, Mn, Zn, Pb, Cr, and Fe ions in water and improve the quality of the water. Kry23-DM cross-linker (1,4,7,13,16-Pentaoxa-10,19 diazo cyclohexene icosane di methacrylate) was synthesized using Kry23 and methacryloyl chloride. The characterization of the molecules was done by FTIR, TGA, DSC, and SEM techniques. The effects of parameters such as pH, concentration, and the metal ion interaction on the heavy metal ions uptaking of SAP was investigated. It was observed that P (AA/Kry23-DM) SAP has maximum water absorption, and the absorption increases with the pH increase. Adsorption rates and sorption capacity, desorption ratios, competitive sorption (q_cs_), and distribution coefficient (log D) of P(AA/Kry23-DM) SAP were studied as a function of time and pH with the heavy metal ion concentration. Langmuir and Freundlich isotherms of the P (AA/Kry23-DM) SAP were investigated to verify the metal uptake. Molecular mechanic (MM2), Assisted Model Building with Energy Refinement (AMBER), and optimized potentials for liquid simulations (OPLS) methods. were used in quantum chemical calculations for the conformational analysis of the cross-linker and the SAP. ΔH^0^_f_ calculations of the cross-linker and the superabsorbent were made using Austin Model 1(AM1) method.

## 1. Introduction

Superabsorbents can absorb large amounts of water compared to common water-absorbing materials, whereby the absorbed water can be removed even under pressure. Due to their superior properties, superabsorbents are widely used in health, agriculture, farming applications, biomedical and daily physiological products, separation technology, and wastewater treatment. In addition, some important studies are carried out in this field. Highly absorbing materials based on polyelectrolyte polymers can absorb up to 50 g of body fluid per gram of dry mass. [1,2,3,4,5]. When the soil fertilizer is sufficient, superabsorbent polymers absorb fertilizer. Superabsorbent polymers are known as hydrophilic network structured polymers having hydrophilic functional groups such as, hydroxyl, carboxylic acid, and amines [5,6,7,8]. 

Recently, superabsorbents are capable of absorbing high amounts of water and have been investigated due to their potential applications in communication technology, construction industry, chromatography, water treatment, and agriculture. Because SAP are widely used, the improvement of their high salt and absorbency properties are becoming increasingly important. The synthesis of a new kind of cost-efficient superabsorbent with excellent swelling properties has drawn much attention in recent years. Li et al. synthesized a kind of poly(acrylic acid)/attapulgite superabsorbent [9,10,11,12,13,14,15]. 

The presence of heavy metals in water is a source of great concern and fear due to aquatic life, human life, and their toxicity to the environment [16,17,18]. For example, lead acting as a general metabolic poison and enzyme inhibitor can lead to mental retardation and brain damage, especially in children. When excess copper builds up in the liver, it causes liver damage [19]. Many researchers have investigated cross-linked polymers synthesized from vinyl monomers such as cross-linked polyacrylamide [20,21], poly(acrylic acid-co-acrylamide) [22], poly(*N*-vinyl imidazole) [23], poly(acryl-amide-maleic acid [24], poly(styrene sulfonic acid-co-maleic acid) [25], and other synthesized polymers [26,27,28,29,30,31].

A force field family of molecular dynamics is Assisted Model Building with Energy Refinement (AMBER) and optimized potentials for liquid simulations (OPLS). The commonly used molecular mechanical energy functions such as AMBER and OPLS were originally parameterized with experimental and quantum chemistry data obtained from small molecules and then compared with experimental observations such as intermolecular energies in gas phase, solution phase densities, and heat of vaporization. To enable the use of the AMBER force field, it is necessary to have parameter values of the force field (e.g., force constants, equilibrium bond lengths and angles, and loads). The parameter set is quite numerous and is described in detail in the AMBER software manual. Each set of parameters that provides parameters for specific types of molecules has a name, and MM2 or force field methods use classical type models to estimate the energy of a molecule as a function of its conformation. One of the semiempirical methods of computational chemistry for quantum computation of molecular electronic structure is AM1 [32,33].

In the present study, because of the presence of carboxylic acid moieties, poly-AA of the superabsorbent having a unique ability to form stable complexes with divalent metal ions, we intended to prepare the SAP from AA by solution polymerization using Kry23-DM cross-linker. We also aimed to investigate the sorption capacity of cross-linked P(AA/Kry23-DM) SAP for the successful separation of heavy metal ions. Then, the adsorption rates and the sorption capacity, the desorption ratios, the competitive sorption (q_cs_), and the distribution coefficient (log D) values of P(AA/Kry23-DM) SAP were investigated over time for Mn^2+^, Ni^2+^, Cu^2+^, Zn^2+^, Pb^2+^, Cd^2+^, Cr^3+^, Co^3+^, and Fe^3+^ heavy metal ions at different pHs; the swelling ratios (q_v_) in water at different pHs were also investigated. Langmuir and Freundlich adsorption isotherms were used to confirm metal uptake data. Langmuir and Freundlich adsorption isotherms were used to validate metal retention data. When 1 N HCl was used as the eluent, it was observed that the removal of all metal ions sorbed of the SAP was high (between 69.8% and 85.6%). As a result, it was found, in the regeneration test results, that the absorber can be regenerated repeatedly. The equilibrium binding capacities of the Mn^2+^, Ni^2+^, Cu^2+^, Zn^2+^, Pb^2+^, Cd^2+^, Cr^3+^, Co^3+^, and Fe^3+^ heavy metal ions by P (AA/Kry23-DM) SAP were 1.4614, 1.4641, 1.4730, 0.8063, 1.4807, 0.7818, 1.4472, 1.4746, and 1.4640 mg/g dry superabsorbent, respectively. The cavity of the cross-linker and the superabsorber was calculated using semiempirical methods to examine whether the ion diameters were suitable for the cavity of the SAP.

## 2. Materials and Methods

### 2.1. Materials

Kryptofix 23 was supplied from Merck company (Kenilworth, NJ, USA). Acrylic acid (AA), triethylamine (Et_3_N), ethylenediaminetetraacetic acid (EDTA), hydroquinone, methacryloyl chloride, sodium persulfate, tertbutyl hydroperoxide and dichloromethane (CH_2_Cl_2_), sodium thiosulfate, ammonium acetate/ammonia (CH_3_COONH_4_/NH_3_) tampon, and the metal chloride reagents of analytical purity (Mn^2+^, Ni^2+^, Cu^2+^, Zn^2+^, Pb^2+^, Cd^2+^, Cr^3+^, Co^3+^, and Fe^3+^ heavy metal ions) were purchased from Aldrich (Sigma Aldrich, St. Louis, MO, USA). The solutions of metal ions were prepared from chloride salts of metals at 1000 mg·L^−1^ concentration in 0.5 M HNO_3_. Solutions composed of metal ions at mg·L^−1^ concentration were prepared freshly by diluting from concentrated metal ion solutions. AA was 99% pure and cleaned by vacuum distillation before use. Hydroquinone (≥99%) was dissolved in dilute hydrochloric acid at low temperature and was crystallized. EDTA (99.5%), Et_3_N (99.5%), Na_2_S_2_O_3_ (99%), Na_2_S_2_O_8_ (98%), tert-butyl hydroperoxide solution 70% in H_2_O, methacryloyl chloride (>98%) (HPLC grade), CH_2_Cl_2_ (HPLC grade), and petroleum ether (95%) were Fluka products, and the purification was not done. Technical CHCl_3_ was left for 1 day in CaCl_2_ and purified by electronic distillation apparatus. The pHs have been set with CH_3_COONH_4_/NH_3_ buffer prepared from CH_3_COONH_4_ reagent grade (≥98%) and ammonium hydroxide solution (28.0–30.0% NH_3_) compounds.

### 2.2. Characterization

#### 2.2.1. Fourier Transform Infrared (FTIR) Spectra

Perkin-Elmer 1600 instrument (Waltham, MA, USA) was used to receive FTIR spectra of SAPs and the cross-linking molecules. The wave number range of the FTIR instrument was 580–4000 cm^−1^, and the scan speed was 4 times per second. The wave number range was 580–4000 cm^−1^ and the scan rate was 4 times per second.

The spectra were taken at room temperature. The 20 mg of SAP or the cross-linker was mixed with 100 mg of dry KBr and pressed to the discs with a pressure of 100 kg·cm^−2^.

#### 2.2.2. ^1^H NMR Spectra 

^1^H NMR spectra of the cross-linker was taken by Varian/Mercury-200 NMR spectrometer (Varian, Yamton, UK). The internal standard was tetramethylsilane and solvent CDCI_3_.

#### 2.2.3. Atomic Absorption Spectrometry (AAS)

AA-929 model AAS (Unicam, Middlesex, UK) was used in measuring of the absorbances of Mn^2+^, Ni^2+^, Cu^2+^, Zn^2+^, Pb^2+^, Cd^2+^, Cr^3+^, Co^3+^, and Fe^3+^ heavy metal ions in the solutions. The spectrometer has 10.0 cm of air/acetylene burner head (5.0 cm of N_2_O/acetylene burner head), 13 mm burner height, and 11.00 L·min^−1^ air flow rate. The spectrometer has a 5 mA lamp and deuterium background correction used in the determination of metal ion concentration. The instrumental parameters of single element hollow cathode lamps were selected as recommended by the manufacturer. The wavelengths selected for the analyte determinations were 240.7, 232.0, 324.8, 228.8, 279.5, 213.9, 217.0, 357.9, and 248.3 nm for Co, Ni, Cu, Cd, Mn, Zn, Pb, Cr, and Fe ions, respectively.

#### 2.2.4. Thermal Gravimetric Analysis (TGA)

The Seiko II Exstar 6000 analyzer (Seiko, Tokyo, Japan) was used to take thermograms of SAP.

TGA measurements of the SAPs were taken in the temperature range of 50–600 °C, at a heating rate of 20 °C/min and under the atmosphere of N_2_ (200 mL/min).

#### 2.2.5. Differential Scanning Calorimeter (DSC)

DSC-141 (Seteram, Caluire, France) was used to examine the thermal properties of the SAP. The thermal properties were investigated at a heating and cooling rate of 10 °C/min under a nitrogen flow of 200 mL/min. The heat exchange was recorded in a cell by loading about 3.8 mg polymer and during heating and cooling cycles.

#### 2.2.6. Scanning Electron Microscopy (SEM)

Scanning electron micrographs of SAPs were taken by JEOL JXA-840 brand electron microscopy (JEOL, Tokyo, Japan). The SAPs were broken after freezing under liquid nitrogen. SEM measurements were performed at 10 kV. The resolution in high vacuum, low vacuum, and ESEM is 3.0 nm at 30 kV. Standard detectors were ETD, low-vacuum SED (LVD), gaseous SED for ESEM mode (GSED), and IR camera. Electron images were recorded directly on a Polaroid film from the cathode ray tube. The magnification in the measurements was up to 3000× and up to 1000×.

## 3. Synthesis

### 3.1. Synthesis of Kry23-DM

Kry23-DM cross-linker was synthesized according to [33]. Kryptofix 23 (Kry23, 1 g, 3 mmol) and 10 mL of CH_2_Cl_2_ were placed in a 100 mL two-necked baloon tied on Schlenk system and was stirred until Kry23 was dissolved. After Kry23 was dissolved, hydroquinone (0.2 g) and Et_3_N (0.85 mL) were added and stirred at room temperature for 1 h. After additions, the flask was immersed in an ice bath. 

Methacryloyl chloride (0.75 mL, 8 mmol) diluted with 2 mL of CH_2_Cl_2_ was dropped to the reaction mixture with a drop funnel and was stirred at 0 °C for 8 h. Et_3_N salts were precipitated with an excess of petroleum ether at the end of the reaction time and then removed by filtration. The reaction solution was purged using a basic alumina column. The resulting product was dried in a vacuum oven at 20 °C for 24 h. The Kry23-DM cross-linker molecule was obtained as a viscous fluid of red-brown color. The reaction path of the Kry23-DM cross-linker is contained in Scheme 1. Figure 1A and Figure 2 show the FTIR and NMR spectra, respectively. Figure 3 shows the optimized geometries of the Kry23-DM cross-linker according to the MM2 method.

### 3.2. Synthesis of the P (AA/Kry23-DM) SAP 

Precisely, 100 g of AA, 500 g of deionized water, 0.1 g of EDTA, and 1 g of Kry23-DM cross-linker were put in a two-necked flask tied on Schlenk system in N_2_ gas medium and was degassed three times. After the reaction solution was stirred at room temperature for 30 min, 3 mL of sodium persulfate solution (0.5 g/4.5 mL deionized water), 1.5 mL of tert-butyl hydroperoxide solution (50 mg/5 mL deionized water), and 4 mL of sodium thiosulfate solution (0.5 g/4.5 mL deionized water) were slowly added into the reaction solution and the flask was placed in an oil bath at 60 °C. When the reaction mixture began to gel, it was terminated by pouring into ice water. The cross-linked P (AA/Kry23-DM) SAP gel was decanted, cleaned three times with deionized water, then dried under vacuum at 100 °C until its weight was stable. The cross-linked P (AA/Kry23-DM) SAP was obtained as a yellowish solid. The SAP is not dissolved in solvents such as methanol, benzene, chloroform, tetrahydrofuran, toluene water, or acetone. Scheme 1 shows the reaction mechanism of P (AA/Kry23-DM) SAP. Figure 1B, Figure 4, Figure 5 and Figure 6 show results of FTIR, TGA, DSC, and SEM analyses, respectively. 

### 3.3. Computational Method

All calculations of the superabsorbent and the cross-linker were performed using HyperChem 8.0 program [34] and Gaussian 03W pocket program. The conformational analysis of the Kry23-DM cross-linker and the SAP were made using MM2(MM+), AMBER, and OPLS methods [35]. The conformation analyses of the cross-linker and the SAP (for n = 8, 20) were achieved by the HyperChem 8.0 program parameters.

The full geometric optimizations of the cross-linker and SAPs were performed using the Polak–Ribiere conjugate gradient method until an RMS gradient of 0.0001 kcal (Å mol)^−1^ was reached.

The defaults of the HyperChem 8.0 program are force field options in the AMBER and OPLS methods. Therefore, for molecular mechanical calculations, the AMBER force field used is characterized by a dielectric constant with a distance of 1.1–4. The electrostatic and van der Waals scale factors are 0.833 and 0.5, respectively. The OPLS force field used is characterized by a dielectric constant with a distance scaled with 1 and 1–4. The electrostatic and van der Waals scale factors of the OPLS force field are 0.5 and 0.125, respectively. In addition, the quantum chemical calculations were investigated with Polak–Ribiere (combined gradient). The algorithm and RMS gradient of Polak–Ribiere is a semiempirical method of RHF/AM1 with a convergence limit of 0.0001 kcal mol^−1^ and 0.001 kcal (Å mol^−1^), respectively. 

The cavity radius of the SAP was calculated as 6.8928 Å from SAP’s optimized geometric structure [36,37,38,39,40,41,42]. Figure 7 shows the optimized geometry of the P (AA/Kry23-DM) SAP according to MM2 method. The calculated heat of formation (Δ_f_H^0^), strain energy (E_str_), internal rotational angle (φ), the bond length (d), the lengths of the repeating unit of polymer chain (ℓ), and internal barrier energy (ΔE) of the stable conformations of the cross-linker and the superabsorbent are shown in Table 1.

### 3.4. Sorption Experiments of the P (AA/Kry23-DM) SAP

#### 3.4.1. Sorption Rate

To investigate the sorption rate of the SAPs, 10 mL of the stock solution containing 15 ppm of heavy metal ions (Mn^2+^, Ni^2+^, Cu^2+^, Zn^2+^, Pb^2+^, Cd^2+^, Cr^3+^, Co^3+^, and Fe^3+^) were prepared individually using deionized water. pHs of the solutions were brought to 7 with NH_4_CH_3_COO/NH_3_ buffer, and the volume of the solution was recorded. Then, 100 mg of P (AA/Kry23-DM) SAP was added to the solutions whose pH was adjusted to 7 and was shaken in a mixer with a speed of 160 rpm at 25 °C. The absorption values of the solutions from 15 to 480 min were measured using AAS, and C values were calculated. With the help of Equation (1), the metal ion concentrations absorbed per unit mass of the superabsorbent copolymer were calculated. The sorption rate results and graphs obtained against different adsorption times of heavy metal ions are given in Table 2 and Figure 8.
(1)$$q=\;\frac{\left(C_{0}-C\right)V}{W}$$

In this equation, q is amount of the metal ion sorbed by the SAP (1 mg/1 g). C_0_ is concentration of the heavy metal ions in initial solution (mg/mL). C is concentration of the metal ions remaining in solution after the sorption time (mg/mL). V is the solution volume (mL) and W is the SAP weight (g) [43].

#### 3.4.2. Sorption Capacity

The sorption capacity is known to be an important factor in determining the amount of superabsorbent required to remove of a particular heavy metal ion from the solution completely. The sorption capacity of SAPs was investigated for Mn^2+^, Ni^2+^, Cu^2+^, Zn^2+^, Pb^2+^, Cd^2+^, Cr^3+^, Co^3+^, and Fe^3+^ heavy metal ions. Precisely, 10 mL solutions with 15, 20, 25, and 30 ppm concentration of the heavy metal ions were prepared, and the pH of the solutions was adjusted to 7 with buffer solution. Specifically, 100 mg P (AA/Kry23-DM) SAP was added separately to the metal ion solutions. The tubes containing the SAPs and metal ions were stirred in a shaker at 160 rpm shaking rate at 25 °C for 8 h. The superabsorbent copolymers were filtered out of solutions containing metal ions after 8 h. Afterwards, the absorbances of the metal ions before the sorption and the residual metal ions in the solutions were measured using AAS, and the concentrations of the metal ions were calculated. The amount of q of the P (AA/Kry23-DM) SAP superabsorbent copolymer was calculated using Equation (1) for each metal ion. The results and the sorption graphics are shown in Table 3 and Figure 9. The calculated *R*^2^ values are as follows: Mn^2+^, 0.9995; Ni^2+^, 1.000; Cu^2+^, 0.9999; Zn^2+^, 0.9896; Pb^2+^, 0.9989; Cd^2+^, 0.8285; Cr^3+^, 0.9659; Co^3+^, 0.9998; and Fe^3+^, 0.9999.

#### 3.4.3. pH Effect on Sorption Capacity

The solutions containing the SAP and the metal ions were shaken at 25 °C for 8 h. The agitation rate was 160 rpm. After shaking, the SAP was checked and the metal ion amount remaining in the solution was determined.

The sorption experiments were repeated three times to see the accuracy of the results. The absorbance of the solutions was measured using AAS. After the C values were determined, the q value of SAP was calculated by Equation (1). Table 4 and Figure 10 show the pH effect on the sorption capacity.

#### 3.4.4. Competitive Sorption

pHs of the solutions containing all of Cu^2+^, Zn^2+^, Pb^2^, Cd^2+^, and Fe^3+^ ions were adjusted to 3, 7, and 9 for the competitive sorption experiments. The concentration of each metal ion in competing adsorption solutions was 15 ppm, and a volume of 10 mL was taken from these solutions. Precisely, 100 mg of P (AA/Kry23-DM) SAP was added to the solutions. The solutions were stirred at 25 °C for 8 h (agitation speed: 160 rpm). C value was calculated using AAS measurements, and then, q_cs_ was calculated with Equation (1) for Cu^2+^, Zn^2+^, Pb^2+^, Cd^2+^, and Fe^3+^ cations. The results and q_cs_ graphic are given in Table 5 and in Figure 11. The calculated R^2^ values are Cu^2+^, 0.5637; Zn^2+^, 0.7159; Pb^2+^, 0.7089; Cd^2+^, 0.7499; Fe^3+^, 0.9647.

#### 3.4.5. Distribution Ratio (log D)

Log D values [44] of P (AA/Kry23-DM) SAP were calculated from the values in Table 4 according to Equation (2) at pH values of 1, 3, 5, 7, and 9. The experimental data and log D graphics are given up in Table 6 and Figure 12. Table 7 shows log D/pH ratios of the heavy metal ions on P (AA/Kry23-DM) SAP (at pH > 3).
(2)$$\log D\;=\;\log\frac{\mathrm{Amount}\;\mathrm{of}\;\mathrm{the}\;\mathrm{cation}\;\mathrm{sorbed}\;\mathrm{by}\;\mathrm{SAP}\left(\mathrm{mmol}/g\right)}{\mathrm{Amount}\;\mathrm{of}\;\mathrm{the}\;\mathrm{cation}\;\mathrm{in}\;\mathrm{the}\;\mathrm{aqueous}\;\mathrm{solution}\left(\mathrm{mmoL}/\mathrm{mL}\right)}$$

#### 3.4.6. Swelling Ratio, q_v_

q_v_ value of SAP was performed at pH 1, 3, 5, 7, and 9. The amount of deionized water was 50 mL, and 0.1 g of P (AA/Kry23-DM) SAP was shaken in a shaker at room temperature for 24 h. q_v_ values of P (AA/Kry23-DM) SAP were calculated using Equation (3):(3)$$q_{v}=\frac{V_{\mathrm{dry}\;\mathrm{polymer}}+V_{\mathrm{solvent}}}{V_{\mathrm{dry}\;\mathrm{polymer}}}=\frac{V_{\mathrm{swollen}\;\mathrm{polymer}}}{V_{\mathrm{dry}\;\mathrm{polymer}}}$$

In the q_v_ equation, the dry polymer volume is V_dry polymer_, the solvent volume is V_solvent_ and the swollen polymer volume is V_swollen polymer_. The polymer density was 1.2442 g/mL, and the water density was taken as 0.9958 g/mL. Table 8 and Figure 13 show the experimental values and the q_v_ graph. The q_v_ values of SAP are 4.74 (at pH 1), 5.10 (pH = 3), 6.16 (pH = 5), 7.00 (pH = 7), and 38.98 (pH = 9) as shown in Table 8. In addition, Figure 13 shows q_v_ graph against different pHs of the SAP in water.

In addition, swelling experiments were also carried out with chloroform, toluene, benzene, and absolute ethyl alcohol solvents, apart from deionized water. As a result, it was found that the cross-linked SAP did not swell in any solvent other than water.

#### 3.4.7. Desorption

The desorption experiments were performed to investigate the stability and repeated use of P (AA/Kry23-DM) SAP. The heavy metal ion solution pHs were brought to 7 with NH_4_CH_3_COO/NH_3_ buffer, and the volume of solution was recorded. Precisely, 100 mg superabsorbent and 10 mL of 30 ppm metal ion containing solution were shaken at 25 °C for 4 h. The agitation rate was 160 rpm. The desorption ability of P (AA/Kry23-DM) SAP was investigated for Mn^2+^, Ni^2+^, Cu^2+^, Zn^2+^, Pb^2+^, Cd^2+^, Cr^3+^, Co^3+^, and Fe^3+^ cations. The SAP beads sorbing the metal ions were eluted from the solution at the end of sorption time and dried in vacuum oven at 25 °C. Subsequently, the SAPs were mixed with 10 mL of 0.1 M HNO_3_ solution for 4 h at 25 °C for desorption of metal ions sorbed by the SAPs. To calculate desorbed metal ion concentrations in HNO_3_ solution, absorbances were measured with AAS. Desorption rate (%) values were calculated using Equation (4).
(4)$$\mathrm{Desorption}\;\mathrm{ratio}\left(\%\right)=\;\frac{\mathrm{Amount}\;\mathrm{of}\;\mathrm{the}\;\mathrm{cation}\;\mathrm{desorbed}\left(\mathrm{mg}\right)}{\mathrm{Amount}\;\mathrm{of}\;\mathrm{the}\;\mathrm{cation}\;\mathrm{sorbed}\left(\mathrm{mg}\right)}\;\times 100$$

The desorption ratios of the metal ions from the P (AA/Kry23-DM) SAP^a^ and ionic diameters^b^ of the metal ions (ID) [45] are shown in Table 9 (a is the cavity of the superabsorbent copolymer and is 3.5022 Å (average) [46], whereas b is diameter of H_2_O and is equal to 2.82 Å) [46].

#### 3.4.8. Freundlich and Langmuir Isotherm Applications

The adsorption isotherm studies were carried out by shaking 100 mg P (AA/Kry23-DM) SAP and the solutions in different concentration of Mn^2+^, Ni^2+^, Cu^2+^, Zn^2+^, Pb^2+^, Cd^2+^, Cr^3+^, Co^3+^, and Fe^3+^ ions for 24 h at 25 °C. The SAPs and 10 mL of 15, 20, 25, and 30 ppm solutions containing metal ions were agitated in a shaker with an agitation speed of 160 rpm. After shaking period, the solution was filtered and the metal ion concentration in the filtrate was calculated using the absorbance values of AAS. The sorption isotherms of heavy metal ions were investigated using Langmuir and Freundlich isotherms. Freundlich and Langmuir isotherms of the metal ions on P (AA/Kry23-DM) SAP were calculated with Equations (5) and (6) using the values in Table 3. Table 10 shows the isotherm results and the calculated parameters.

Equation (5) is the Freundlich isotherm equation.
q = k C^1/n^(5)

In Freundlich equation, the metal ion adsorption amount of 1 g of P (AA/Kry23-DM) SAP is q (mmol/g) in the equilibrium. C is the metal ion concentration in the solution in the equilibrium (mmol/L), k is Freundlich constant, and 1/n are empirical parameters [33,47,48]. 

## 4. Results and Discussion

### 4.1. Synthesis of Kry23-DM Cross-Linker and P (AA/Kry23-DM) SAP

Synthesis of Kry23-DM cross-linker and P (AA/Kry23-DM) SAP were synthesized according to Synthesis of Kry23-DM section and Synthesis of the P (AA/Kry23-DM) SAP section. The cross-linked P (AA/Kry23-DM) SAP was prepared by solution polymerization/cross-linking. The synthesized polymer was characterized and evaluated as a sorbent for the removal of toxic metal ions (Mn^2+^, Ni^2+^, Cu^2+^, Zn^2+^, Pb^2+^, Cd^2+^, Cr^3+^, Co^3+^, and Fe^3+^) from aqueous solutions.

### 4.2. Characterization of Kry23-DM Cross-Linker

The characterization of Kry23-DM cross-linker was achieved by FTIR and ^1^H NMR techniques. FTIR spectrum of Kry23-DM cross-linker in Figure 1A shows the characteristic peaks at 1614 cm^−1^ for C=C groups, at 1762 cm^−1^ for C=O groups, at 2870–2923 cm^−1^ for –CH_3_ and –CH_2_ groups, and at 1270 cm^−1^ for –C–N group. The characterization of Kry23-DM cross-linker was achieved by ^1^H NMR technique. Figure 2 shows ^1^H NMR spectrum of Kry23-DM. (s, 6H, –CH_3_), (m, 8H, N–CH_2_), (m, 2OH, O–CH_2_), and (d, 4H, =CH_2_) group peaks of Kry23-DM cross-linker are at 1.98, 3.02–3.18, 3.51–3.78, and 4.98–5.17 ppm, respectively.

### 4.3. Characterization of P (AA/Kry23-DM) SAP

FTIR spectrum of the cross-linked P (AA/Kry23-DM) SAP in Figure 1B shows the characteristic peaks at 1730 cm^−1^ for C=O groups of PAA chains of SAP and Kry23-DM cross-linker, at 1174 cm^−1^ for C-O groups of Kry23-DM cross-linker, at 2860–2934 cm^−^^1^ for CH, –CH_2_, and –CH_3_ groups of AA, and Kry23-DM cross-linker, at 3408 cm^−^^1^ for OH peaks of AA, and at 1270 cm^−1^ for –C–N group of Kry23-DM cross-linker. Since the P (AA/Kry23-DM) SAP was cross-linked, it was insoluble in any solvent. Thus, ^1^H NMR spectrum of the superabsorbent copolymer could not be obtained.

The thermal behavior of the P (AA/Kry23-DM) SAP was investigated, and the thermogravimetric traces of the formulations prepared are shown in Figure 4. It is apparent from the Figure 4 that P (AA/Kry23-DM) SAP loses weight in four decomposition steps. The thermal stability was evaluated by comparing the T_i_ (initial temperature of degradation), T_max_ (temperature of maximum rate of weight loss), T_f_ (final temperature of degradation), and char yield at 536 °C, which are presented in Table 11. 

Figure 4 shows the TGA and DTG curves of P (AA/Kry23-DM) SAP. It is apparent from the Figure that this polymer loses weight in four decomposition steps. The first step occurs between 30 and 156 °C and with a weight loss of 3.9 wt% from adjacent carboxylic moieties to form anhydride.

The second step starts at 160 °C and ends at 258 °C with a maximum decomposition rate at 258 °C and with a weight loss of 19.0–16.0 wt% from the loss of CO_2_, which coincides with the decomposition of the main chain [49,50]. Then, the third step starts with a maximum decomposition at 417 °C and with a weight loss of 39.9 wt% and then followed by the last step with a maximum decomposition rate at 536.7 °C and with a weight loss of 85.3 wt%. The degradation of the P (AA/Kry23-DM) SAP chain began at 417 °C and % char yield remained at 14.7 wt% at over the 536 °C. The results reveal that cross-linking does not lead to any significant change in the thermal degradation behavior.

The DSC thermogram of the superabsorbent is located in Figure 5. In this thermogram, there are three endothermic peaks at 92.43, 252.55, and 446.84 °C. Two exothermic peaks can be seen at 427.25 and 479.38 °C in the thermograms. The first peak in the DSC thermogram corresponds to endothermic transition and can be attributed to water loss. The second peak at 252.55 °C corresponds to the endothermic transition and can be attributable to PAA anhydride formation. The third exothermic peak at 427.24 °C and the fourth exothermic peak at 478.37 °C can be attributable to the thermal decomposition of PAA anhydride. TGA and DSC obtained results are in good agreement with the data reported in the literature [51,52], especially those obtained by Cardenas et al. [50]. 

The surface appearance and structure of superabsorbent were observed using scanning electron microscopy. SEM photograph of the superabsorbent’s dense and smooth surface of the backbone is shown in Figure 6. As is evident from Figure 6, cross-linking of PAA chains with the cross-linker resulted in homogeneous but fractured topography, which appears nonporous in topography (a, b, and c in 100, 20, and 200 μm magnifications, respectively). The surface is smooth and homogeneous (d and e in 100 and 100 μm magnifications, respectively) for cross-linked P (AA/Kry23-DM) SAP. SEM images show that superabsorbent surface is porous, branched, and rough (f, g, h, j in 100, 20, 100, and 100 μm magnifications, respectively). Kry23-DM acts as a bridge (cross-linker) between polymer chains in the cross-linked structure and forms a microporous structure by bonding PAA chains together.

### 4.4. Computational Method

Because of the ability to predict the key physical and chemical properties of polymeric materials from their repeat-unit structure, chain-length architecture prior to synthesis is of great value for the design of polymer-based chemical products, with new functionalities and improved performance, we calculated Δ_f_H^0^, E_str_, the lengths of the repeating unit of SAP chain (ℓ), and the cavity of Kry23-DM cross-linker and P (AA/Kry23-DM) SAP. Figure 7 shows the optimized geometries of P (AA/Kry23-DM) SAP according to MM2 method. The full geometry optimizations of the P (AA/Kry23-DM) SAP chain (n = 8, 20) were performed with the AM1, MM2, AMBER, and OPLS methods and the heat of formations, their strain energies and the lengths of the repeating unit of SAP chain (ℓ) were calculated. E_str_ (kcal mol^−1^), Δ_f_H^0^ (kcal mol^−1^), φ (°), and d (Å) of the stable conformations of Kry23-DM cross-linker and the superabsorbents are given in Table 1.

The semiempirical AM1 method [31] was used to calculate the heat of formation (Δ_f_H^0^) of Kry23-DM cross-linker and P (AA/Kry23-DM) SAP. The most stable conformation of the Kry23-DM cross-linker was obtained by complete geometric optimization using MM2 method. Figure 7 shows the optimized geometries of the most stable conformation of the Kry23-DM cross-linker. The torsion angle formed by the plane of the macrocyclic ring of the polymer chains in the stable conformation is about 81°. However, it has also been found to be inclined toward the inner region of the macrocyclic cavity of the nonbonding electron pairs of heteroatoms in the stable conformations. For the P (AA/Kry23-DM), the internal cavity radii of the macrocyclic fragment, R was calculated from the distance values between heteroatoms, and R is found as 3.4464 Å. The cavity radii of 18-crown−6 macrocyclic were previously given as 2.6–3.2 Å in the literature [45]. The radii of the internal cavity of the macrocyclic fragment for the cross-linked poly(AA-co-Kry-22-DM) superabsorbent copolymer has been calculated as R = 3.22 Å [29]. The radii of the internal cavity of P (AA/Kry23-DM) SAP is bigger than poly(AA-co-Kry-22-DM) superabsorbent copolymer.

According to the results of the molecular mechanics (MM2, AMBER, and OPLS), the 1–2–3–4 internal rotation angle (dihedral) (φ) for the cross-linker and the polymers in the stable conformations is 173.0380°, 175.2091°, and 174.394° (for the cross-linker, the superabsorbents with n = 8 and 20) and the internal rotation angles are almost equal to 180°. As shown in Table 2, the lengths of the repeating unit were calculated and found as 7.8315, 7.7733, and 7.3945 Å (for n = 2) and 27.8679, 24.7956, and 26.2141 Å (for n = 8) for the superabsorbents with MM2, AMBER, OPLS methods, respectively. According to OPLS, the superabsorbent chain (n = 8 and 20) had the lowest strain energy (Table 1). The internal rotation barrier energies (ΔE) were calculated (Table 1), and it was determined that the superabsorbents calculated by OPLS had the lowest ΔE. Thus, the superabsorbent prepared in the liquid reaction medium simulations should be more flexible than gaseous reaction medium simulations.

### 4.5. Sorption Experiments of the P (AA/Kry23-DM) SAP

#### 4.5.1. Sorption Rate

Adsorption rates of P (AA/Kry23-DM) SAP were investigated as a function of time with the metal ion concentration. The initial concentrations of the metal ions in the aqueous phase were taken as 15 ppm. The pH of the medium was set to pH = 7. At different times, concentrations of metal ions in samples taken from the aqueous solution were measured at atomic absorption spectrometry. Adsorption rates were determined by plotting milligram amounts of metal ions adsorbed by 1 g P (AA/Kry23-DM) SAP against time. The adsorption rates of heavy metal ions are shown in Figure 8 and Table 2.

As observed from these figures, the initial adsorption rate is high and a straight line is reached after a certain time. This straight line shows that adsorption equilibrium is reached. This equilibrium time varies between 66′ and 107′ for working metal ions. The amount of adsorbed metal ion is also increasing. The high swelling rate at basic medium pHs increases the interaction of metal ions with P (AA/Kry23-DM) SAP.

Adsorption rates as a function of time with metal ions of P (AA/Kry23-DM) SAP were investigated. The initial concentrations of the metal ions in the aqueous phase adsorption of metal ions is particularly rapid, especially when the concentration of metal ions is high. This is due to the repulsive force, which is abundant in the high metal ion concentrations between the aqueous phase and the solid phase (i.e., superabsorbent particles). The adsorption rates of metal ions were found to be 107, 67, 66, 70, 72, 68, 73, 71, and 69 min for Mn^2+^, Ni^2+^, Cu^2+^, Zn^2+^, Pb^2+^, Cd^2+^, Cr^3+^, Co^3+^, and Fe^3+^, respectively.

Although the adsorption rates of metal ions by the P (AA/Kry23-DM) SAP particles are very close to each other, the P (AA/Kry23-DM) SAP particles show a greater affinity for Cu ions and are adsorbed more rapidly than other ions. The order of adsorption rate of metal ions by the P (AA/Kry23-DM) SAP is as follows:Cu^2+^ > Ni^2+^ > Cd^2+^ > Fe^3+^ > Zn^2+^ > Co^3+^ > Pb^2+^ > Cr^3+^ > Mn^2+^

Although the sorption rate for Pb^2+^ cation was the highest, because the affinity of the poly(AA-co-Kry22-DM) superabsorbent copolymer to Pb^2+^ cation was the highest, the order of the sorption rates of the P (AA/Kry23-DM) SAP for the metal ions is Pb^2+^ > Cu^2+^ > Cr^3+^ = Cd^2+^ = Zn^2+^ > Ni^2+^ >Mn^2+^ [33]. The P (AA/Kry23-DM) SAP particles has observed a greater affinity for Cu^2+^ ions, and Cu^2+^ > Ni^2+^ > Cd^2+^ > Fe^3+^ > Zn^2+^ > Co^3+^ > Pb^2+^ > Cr^3+^ > Mn^2+^ is the adsorption rate order in this research. Although the sorption rates of Cr^3+^, Cd^2+^, and Zn^2+^ on the poly(AA-co-Kry22-DM) superabsorbent copolymer [33] are equal, the order of the sorption rate of the P (AA/Kry23-DM) SAP is Cd^2+^ > Zn^2+^ > Cr^3+^. The cavity increasing of P (AA/Kry23-DM) SAP was affected by adsorption rates of the metal ions.

#### 4.5.2. Sorption Capacity

The adsorption capacity of P (AA/Kry23-DM) SAP was expressed as a function of the initial concentrations of metal ions in the aqueous phase. Adsorption capacities of the metal ions adsorbed by 1 g P (AA/Kry23-DM) SAP against the concentrations of metal ions in the aqueous phase are shown in Figure 9 and Table 3. The concentrations of metal ions in the aqueous phase are ranged from 15 to 30 ppm. The adsorption amounts of metal ions are different according to their concentrations. When the results in Figure 9 and Table 3 are compared, adsorption order of metal ions in the concentration range of 25–30 ppm is as follows:Fe^3+^ > Pb^2+^ > Ni^2+^ > Cu^2+^ > Co^3+^ > Cr^3+^ > Mn^2+^ > Zn^2+^ > Cd^2+^

It was observed from the graphics that the adsorption capacity of Fe^3+^ and Pb^2+^ ions was higher than those of other ions. The amounts of heavy metal ions adsorbed by the P (AA/Kry23-DM) SAP particles in solutions containing 30 ppm of metal ions were 2.9531, 2.9487, 2.9345, 2.9332, 2.9246, 2.9190, 2.8863, 1.4037, and 0.9071 as mg/g of the polymer. R^2^ values of Mn^2+^, Ni^2+^, Cu^2+^, Zn^2+^, Pb^2+^, Cd^2+^, Cr^3+^, Co^3+^, and Fe^3+^ are 0.9995, 1.000, 0.9999, 0.9896, 0.9989, 0.8285, 0.9659, 0.9998, and 0.9999, respectively. R^2^ values in regression of the ions show the total variation explained in the regression and also show how good fit the model is. The values changing between 1.000 and 0.8285 of R^2^ shows that sorption capacity of the various ions included in the line are good.

The experimental loading capacity of P (AA/Kry23-DM) SAP was found to be in the same range (Figure 9 and Table 3). The P (AA/Kry23-DM) SAP was observed to exhibit the highest sorption capacity for Fe^3+^, followed by Pb^2+^, Ni^2+^, Cu^2+^, Cr^3+^, Mn^2+^, Zn^2+^, and Cd^2+^. The exceptional chelating ability, especially for Fe^3+^, can be explained on the basis of the high stability constant of the iron complex with carboxylic acids in the entire transition metal series.

The order of the sorption capacities of the poly(AA-co-Kry22- DM) superabsorbent copolymer for the metal ions was found as Pb^2+^ > Cu^2+^ > Cd^2+^ > Cr^3+^ > Zn^2+^ > Mn^2+^ > Ni^2+^ [29]. When the sorption capacity results of the P (AA/Kry23-DM) SAP and the poly(AA-co-Kry22-DM) superabsorbent copolymer were compared, although Pb^2+^ sorption order was the same in the two the superabsorbents, it was found that the order of sorption of the other ions was different. Although the Cd^2+^ ions’ sorption capacity of poly (AA-co-Kry22-DM) superabsorbent copolymer is higher than Cr^3+^, Zn^2+^, Mn^2+^, and Ni^2+^ ions, it has been observed that P (AA/Kry23-DM) SAP has the lowest sorption capacity for Cd^2+^ ions.

#### 4.5.3. pH Effect on Sorption Capacity

It is known that the adsorption of metal ions by specific solvents is dependent on the pH [50,51,52,53,54,55]. The reasons such as being of complexing reagents, hydrolysis and precipitation of metal ions were affected by the concentration of adsorbed metal ions. Metal ions precipitates at higher pH values than these pH values. The manner has been examined in detail by Wang [41], Boomhower [53], Caban [54], and Reed [55]. Thus, we worked at pH 1, 3, 5, 7, and 9 to investigate pH effect on the sorption capacity. 

It is determined that the hydrolysis of the metal ions was initiated at pH = 4.6, 2.6, 2.9, and 6.8 for Cr, Fe, Pb, and Cu ions, respectively. The theoretical and experimental settling curves indicate that the precipitation of metal ions is dependent on the concentration of the metal ions in the medium and that precipitation starts above these pH values. Therefore, we selected 1–9 pHs to study the pH effect on the adsorption of metal ions over P (AA/Kry23-DM) SAP particles in our study. In this group of experiments, the concentration of aqueous solution of heavy metal ions are at 30 ppm. Figure 10 and Table 4 show the pH effect of the metal ion adsorbed by 1 g P (AA/Kry23-DM) SAP against the concentrations of metal ions in the aqueous phase at different pH values. 

Adsorption capacities were found to be lower with pH < 3 than those adsorbed at other in pH media for Cd, Co, Ni, and Cu ions. As can be seen from the figures, the adsorption of heavy metal ions increases with increasing pH and then becomes a flat plateau. This plateau represents the state of equilibrium.

The affinity order of P (AA/Kry23-DM) SAP for metal ions bound is determined as Pb^2+^ > Fe^3+^ > Mn^2+^ > Cr^3+^ > Cd^2+^ > Co^3+^ > Cu^2+^ > Zn^2+^ > Ni^2+^ at pH < 3. The amounts of metal ions adsorbed by 1 g P(AA-co-Kry-23-DM) SAP are 2.4858, 2.9246, 1.5315, 2.7092, 0.9892, 2.6105, 2.8848, and 2.9339 mg for Fe^3+^, Pb^2+^, Ni^2+^, Cu^2+^, Co^3+^, Cr^3+^, Mn^2+^, Zn^2+^, and Cd^2+^ at pH = 9, respectively. In addition, when pH value increases, the sorption capacity of poly(AA-co-Kry22-DM) SAP for Pb^2+^, Cu^2+^, Cr^3+^, Cd^2+^, Mn^2+^, Zn^2+^, and Ni^2+^ cations increase initially, and it was observed that the affinity of the poly(AA-co-Kry22-DM) SAP to the metal ions decreases with an order of Pb^2+^ > Cr^3+^ > Cu^2+^ > Cd^2+^ > Zn^2+^ > Mn^2+^ > Ni^2+^ [33]. The sorption capacity of both SAPs is highest for Pb ions. Mn^2+^ ions have been much more preferred by the P (AA/Kry23-DM) SAP than poly(AA-co-Kry22-DM) SAP. 

#### 4.5.4. Competitive Sorption

The results of competing adsorption are shown in Figure 11 and Table 5 in solutions containing Fe^3+^, Cu^2+^, Zn^2+^, Pb^2+^, and Cd^2+^ ions at pH = 1, 3, 5, 7, and 9. The values changing between 0.5637 and 0.9647 of R^2^ show that competitive sorption of the various ions included in the line are good in Figure 11.

The competitive adsorption quantities of these ions are different from the adsorption of single metal ions. While the competitive adsorption quantitative is 1.4585 mg/g of polymer alone for Fe^3+^ at pH = 7, this amount is found as 1.4707 mg/g of the polymer in the case of competing adsorption.

The adsorption amount alone for Cd^2+^ at pH = 7 was 0.7928 mg/g of polymer, while in the case of competing adsorption, this amount was found to be 0.9940 mg/g of the polymer. Although the adsorption amount for Cu2+ alone was 1.4791 mg/g of polymer at pH = 7, it was found to be 1.0780 mg/g of polymer in the case of competing adsorption at pH = 7. Although the adsorption amount for Pb^2+^ alone was 1.4972 mg/g of polymer, this amount was found to be 1.1418 mg/g of polymer in the case of competing adsorption. Although the amount of adsorbed alone for the Zn^2+^ ion was 0.8318 mg/g of polymer at pH = 7, this amount was found to be 1.0164 mg/g of polymer in the case of competing adsorption. Although the order of the competitive sorption values for the metal ions were Pb^2+^ > Cu^2+^ > Cd^2+^ > Zn^2^^+^ on the poly(AA-co-Kry22-DM) at pH 3, 7, and 9 [33], the order of competitive sorption values for P (AA-co-Kry23-DM) SAP was found as Cu^2+^ > Fe^3+^ > Zn^2+^ > Cd^2+^ > Pb^2+^ at pH = 1, 3, 5, 7, and 9. Although there is not much difference for the Fe3+ ion in either case, it is not the case for other ions. This can be attributed to the interaction of charged ion groups. When ions enter the race together, it is usually not possible to generalize the order of adsorption and to estimate the amount of metal ions adsorbed. As a result, the competitive sorption values (qcs, Table 5) for Cu^2+^, Fe^3+^, Zn^2+^, Cd^2+^, and Pb^2+^ cations on P (AA/Kry23-DM) SAP and the poly(AA-co-Kry22-DM) superabsorbent copolymer were usually smaller than the sorption values (q, Table 3) because of the repulsion between the cations.

#### 4.5.5. The Distribution Coefficients (log D)

The distribution coefficients (log D) of the metal ions in the aqueous phase and solid phase distribution of P (AA/Kry23-DM) SAP at different pHs are shown in Figure 12 and Table 6. The log D values against pH were calculated with Equation (2) using the values in Figure 12 and Table 6 at different pHs. Figure 12 and Table 6 show the relationship between log D and pH. The curve of the logarithm of the Ni ion falls to the negative region at pH = 1 below. The values of metal ions (d log D/pH) are changed from 0.24 to 0.57 for pH > 3 (Table 7). It has been found that the d (log D)/pH value on the poly(AA-co-Kry22-DM) superabsorbent copolymer has changed between 0.14 and 0.74 at pH > 3. These show that the metal ions are adsorbed by P (AA/Kry23-DM) SAP and the poly(AA-co-Kry22-DM) superabsorbent copolymer. The adsorption selectivity of metal ions by the P (AA/Kry23-DM) SAP for pH < 3 is shown in the graph of Figure 12 and Table 7 as follows: Pb^2+^ > Fe^3+^ > Mn^2+^ > Cr^3+^ > Co^3+^ > Cd^2+^ > Cu^2+^ > Zn^2^^+^ > Ni^2+^.

#### 4.5.6. The Swelling Ratios (q_v_)

q_v_ values were calculated according to Equation (2), and q_v_ values are presented in Figure 13 and Table 8. q_v_ values in water of the P (AA/Kry23-DM) SAP were carried out at pH = 1, 3, 5, 7, and 9. Experimental results were substituted in Equation (3) and q_v_ values were calculated. The swelling ratios of P (AA/Kry23-DM) SAP were found to be 4.74, 5.10, 6.16, 7.00, and 38.98 at pH 1, 3, 5, 7, and 9, respectively. The q_v_ values have been found 5.25 at pH = 3; 7.25 at pH = 7, and 10.90 at pH = 9 of the poly (AA-co-Kry22-DM) superabsorbent copolymer [33]. The q_v_ values of the P (AA/Kry23-DM) SAP are approximate to the q_v_ values of the poly (AA-co-Kry22-DM) copolymer except at pH = 9. The q_v_ values of the poly (AA-co-Kry22-DM) superabsorbent copolymer were 5.25 at pH 3, 7.25 at pH 7, and 10.90 at pH 9. The q_v_ values of P (AA/Kry23-DM) SAP are three times bigger than the q_v_ values of the poly (AA-co-Kry22-DM) copolymer. It was observed that the q_v_ values of P (AA/Kry23-DM) SAP increased by the swelling of the superabsorbent with pH increasing and the amount of adsorbed metal ions is also increased. The high swelling ratio at basic medium pHs increases the interaction of metal ions with the P (AA/Kry23-DM) SAP. 

The q_v_ should increase with increasing fixed charges per polymer volume. As the degree of neutralization of AA increases, more carbocyclic groups are ionized and the tendency to expand the chain network with increased electrostatic push increases. As a result, water absorption and gel strength are increased. The SAPs developed can retain water and moisture and function as water reservoirs. The absorption of superabsorbent polymers depends on the strength of the hydrophilic groups, cross-linking density, polymer network behavior and elasticity of the polymer networks, type of solvent, and strength of the external solution, as well as the characteristics of the external solution, etc. [56].

#### 4.5.7. Desorption

For adsorption of metal ions adsorbed by P(AA/Kry23-DM) SAP, the recovery of adsorbed quantities obtained from solutions containing 30 ppm metal ion at pH = 7 in acidic medium (pH = 1) was investigated. As shown in Table 9, the desorption rate of ions except Fe^3+^ (23.7%) is 65.6–85.6%. The desorption rate order of the P (AA/Kry23-DM) SAP is Pb^2+^ > Cd^2+^ > Cu^2+^ > Mn^2+^ > Zn^2+^ > Cr^3+^ > Co^3+^ > Ni^2+^ > Fe^3+^. The desorption rate of ions was found between 65.6% (for Ni^2+^) and 85.6% (for Pb^2+^). The desorption ratios of the P (AA-co-Kry22-DM) superabsorbent copolymer was between 66.4% (for Cr^3+^) and 82.8% (for Pb^2+^). Once the recovery was done, a high recovery was obtained for all the metal ions with 1 N HCl as eluting agent. The desorption rate of both the superabsorbent is close together. The desorption rate of both the superabsorbent is close together. The ion diameters are 2.42, 1.34, 1.44, 1.26, 1.60, 1.40, and 1.82 Å for Pb^2+^, Ni^2+^, Cu^2+^, Cr^3+^, Mn^2+^, Zn^2+^, and Cd^2+^, respectively [42,43,44,45,46]. The ion diameters (Å) of hydrated metal ions in aqueous solution are 1.20, 0.715, 0.62, 0.65, 0.86, 0.74, and 0.96 Å for Pb^2+^, Ni^2+^, Cu^2+^, Cr^3+^, Mn^2+^, Zn^2+^, and Cd^2+^, respectively [46]. The decrease in desorption ratio for the Fe^3+^ may be due to the complexation tendency of the ion. It can also be said about the difference in ion diameters. According to the desorption results obtained, it is observed that the desorption increases as the ion diameter increases.

#### 4.5.8. Adsorption Isotherms 

Langmuir and Freundlich equilibrium isotherm theories are adopted to analyze the experimental data for the metal ion adsorption of the P(AA-co-Kry23-DM) superabsorbent copolymer in this study. Freundlich and Langmuir isotherms of the superabsorbent copolymer were calculated with Equations (5) and (6) using the values in Table 3. The calculated parameters are listed in Table 10. The plot of q versus C for the adsorption of the metal ions onto the superabsorbent is drawn and fitted to the Langmuir and Freundlich isotherms. The values of the Langmuir adsorption constant (K_b_) and the maximum adsorption capacity of the superabsorbent (q), Freundlich constants k and 1/n, and the correlation coefficients (R^2^) for Langmuir and Freundlich model [57,58,59,60] are compared in Table 10. The correlation coefficients (R^2^) of these two models are compared in Table 10. According to Freundlich isotherm, R^2^ values are 0.9448, 0.7922, 0.9485, 0.6939, 0.3465, 0.8698, and 0.9285 for Ni^2+^, Mn^2+^, Zn^2+^, Cd^2+^, Pb^2+^, Cu^2+^ and Co^3+^ ions, respectively. R^2^ values of Langmuir isotherm values are 0.4138, 0.3716, 0.8257, 0.9863, 0.9013, 0.6999, 0.9459) for Ni^2+^ Mn^2+^ Zn^2+^ Cd^2+^ Pb^2+^ Cu^2+^, Co^3+^ ions, respectively. It is obvious that the Freundlich model (0.9448, 0.7922, 0.9485, 0.6939, 0.3465, 0.8698, 0.9285)) yields a much better fit than that of Langmuir model (0.4138, 0.3716, 0.8257, 0.9863, 0.9013, 0.6999, and 0.9459) in Ni^2+^, Mn^2+^, Zn^2+^, Cd^2+^, Pb^2+^, Cu^2+^, and Co^3+^ ions adsorption behavior, respectively. The correlation coefficients (R^2^) of the Langmuir model (0.9599 and 0.5211) are much better fit than the Freundlich model (0.0504 and 0.0926) for Cr^3+^ and Fe^3+^ ions’ adsorption behavior. According to the results, Langmuir isotherm and Freundlich isotherm were in a good agreement with the positive sorption results for Cr^3+^, Mn^2+^, Zn^2+^, Cd^2+^, Pb^2+^, Cu^2+^, Co^3+^, Mn^2+^, and Fe^3+^ cations. It was found that Langmuir isotherm results for the poly(AA-co-Kry22-DM) superabsorbent copolymer were not in a good agreement with the sorption results of Pb^2+^, Cu^2+^, Cr^3+^, Cd^2+^, Mn^2+^, Zn^2+^, and Ni^2+^ cations but Freundlich isotherm is suitable. For Ni^2+^, negative adsorption was observed according to the Langmuir isotherm, whereas in Freundlich isotherm, positive adsorption was observed. Langmuir isotherm shows a negative sorption on the superabsorbent copolymer for Ni^2+^. The concentration of the solute in the superabsorbent copolymer is less than the concentration of the solute remaining in the solution. This behavior called negative sorption is persorption and is displayed by electrolytes especially. The persorption was observed in the calculation made according to the Langmuir isotherm for Ni^2+^ ion. The reasons of persorption is not held to the P (AA/Kry23-DM) SAP with their normal valence bonds of the water molecules and is due to fit into vacant spaces and to attach to the hetero 18-crown-6 rings in the superabsorbent copolymer [46]. This situation can be clearly seen in diapers in children or women. As a result, the first of the major conclusion of this study is that prepared P(AA/Kry23-DM) SAP were able to remove Pb^2+^, Fe^3+^, Mn^2+^, Cr^3+^, Co^3+^, Cd^2+^, Cu^2+^, Zn^2+^, and Ni^2+^ ions from waste water and the second conclusion is that it can be used as a diaper in children or women as a new SAP. 

## 5. Conclusions

In recent years, polymeric adsorbents have been widely used in water treatment because of high water retention capacities and reuse. The researchers have the ability to create porous structures of polymeric adsorbents and to adjust and control the size of these porous structures to the desired size. The production of adsorbents in the purpose suitable property provides the increasing affinity to the superabsorbent polymers. 

This work describes the synthesis, characterization, and analytical evaluation of the cross-linked PAA superabsorbent as a sorbent for the uptake of metal ions like Cr^3+^, Mn^2+^, Zn^2+^, Cd^2+^, Pb^2+^, Cu^2+^, Co^3+^, Mn^2+^, and Fe^3+^ cations from aqueous solutions. The cross-linked PAA superabsorbent is the new polymer as an agent for water purification. In addition, MM2, AMBER, and OPLS and AM1 quantum chemical calculations were used to investigate Δ_f_H^0^, E_str_, ℓ, φ, and d of SAP and the cavity of Kry23-DM cross-linker and P (AA/Kry23-DM) SAP in this study. For P (AA/Kry23-DM), the internal cavity radii of the macrocyclic fragment, R, is found as 3.4464 Å. It was observed from the lowest ΔE values calculated with the OPLS method that it was more flexible in liquid reaction medium simulations.

The synthesized hydrogel exhibits high-metal uptake, and the sorbed metal can be readily eluted by 0.1 N HNO_3_. Once the recovery was done, a high recovery (between 69.8% and 85.6% at 25 °C at 4 h) was obtained for all the metal ions with 1 N HCl as eluting agent. The metal uptake data was found to fit well into both Langmuir and Freundlich adsorption models. The cross-linked superabsorbent was found to be stable over the entire pH range which is soluble in aqueous solutions. The hydrogel was stable and could be reused for at least one time without any loss in its metal uptaking efficiency.

The P(AA/Kry23-DM) SAP is a superabsorbent that it is capable of adsorbing heavy metal ions and analytically reusing it due to its ability to desorb heavy metal ions in acidic media and is also solvent resistant. As the concentration of the heavy metal ion in the solution is increased, the amount of heavy metal ions adsorbed of polymer in the unit amount is also increased at a definite rate [54]. To increase the desorption ratio, the pH of the medium (such as pH < 1) can be lowered by increasing the HNO_3_ concentration. 

The superabsorbent can be used with the purpose of removing the water formed by the moisture of air from the solvents because it is observed that it is insoluble in solvents such as toluene, chloroform, benzene, TFA, acetone, and carbon tetrachloride. In addition, it is possible to find its use in children, hygiene, and adult diapers because of having high water retention properties.

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
