# Peer review of "Synthesis, Characterization of a New Polyacrylic Acid Superabsorbent, Some Heavy Metal Ion Sorption, the Adsorption Isotherms, and Quantum Chemical Investigation"

_materials, 2020, doi:10.3390/ma13194390_

Round 1

Reviewer 1 Report

The authors describe the synthesis of a superabsorbant polymer with Kryptofix23-dimethacrylate as a cross-linker for polyacrylic acid. The content of the work is nearly equivalent to a previous paper of the corresponding author from 2007 (reference 29). The only difference is the the change from Kryptofix22-dimethacrylate to Kryptofix23-dimethacrylate. This means that the crown ether, which could be responsible for the heavy metal ion sorption has been extended by a C2H4O-unit.

As a whole, the manuscript is difficult to read, because of lots of typing errors and inacceptable English language. Only some examples are listed below:

  • Groupssuch
  • SAR
  • Alsynthesized
  • poly (acrylicacid-co-acrylamide
  • acid-comaleic acid
  • were made AM1 method
  • Synthesis of Kry23-DMKry-23 DM
  • Kyr23-DM
  • IR -> FTIR
  • 5 mLof
  • was not dissolved
  • displayed at Table 1
  • Crosslinke
  • r
  • losses weight
  • was began at
  • Char
  • yield were remained
  • crosslinking
  • polimer
  • copolymers was filtered off

Some sentences are very long and therefore difficult to understand.

Furthermore, it is difficult to follow the content, because the figures and tables are far away in the document from its interpretation, so that the reader has to scroll up and down permanently.

Further formal bugs are:

  • Fig 8 is missing.
  • 1+2 look like hand-painted. Please improve quality
  • Scheme 1: Bad aspect ratio
  • The first part in 3.1., 3.2, 3.3, 3.5, 3.6, 3.7, 3.8 belong to the experimental part

In the experimental part are some inaccuracies. Some examples for this are:

  • stirred in the cold: Which temperature?
  • „3 mL of sodium persulfate solution“ and “5 mL of tert-butyl hydroperoxide solution “ are added twice
  • the reaction was terminated: How?
  • adjusted to pH : which pH
  • “SAP was prepared by emulsion polymerization”: I don´t believe this, because in an emulsion polymerization, polymer particles are obtained and the viscosity stays low. In the experimental is written, that the solution is gelling during reaction. Please redefine the type of polymerization.

FTIR of cross-linker and SAP: Is there any evidence, that the cross-linker is built in? Are in the FTIR of SAP any signals, which are characteristic for the cross-linker (e.g. C-N or C=O of amide)?

Overall, the results might be interesting, but the reader has first to read reference 29 to find out, what could be interesting. The authors should consider and discuss in more detail the difference between Kryptofix22-dimethacrylate and Kryptofix23-dimethacrylate in their discussions and in the conclusion. Without this comparison, the paper is just another study of the metal ion sorption capacity of a modified cross-linked polyacrylic acid.

Against the content of the title and the introduction, where the SAP are foreseen for water treatment to remove heavy metal ions, the final conclusion here is: ”it is also possible to find the use area in terms of having high water retention properties in children, hygiene and adult diapers”. For this application much cheaper SAP are on the market for long time. The conclusion should say something about the potential of the “new” polymer as an agent for water purification.

Reviewer 2 Report

In the present work, the authors have synthesized a superabsorbent polymer poly (acrylic acid / cryptofix23-dimethacrylate) by solution polymerization to remove various heavy metal ions in water. FTIR, TGA, DSC and REM methods were used for characterization. The influence of parameters such as pH, concentration and interference of common ions on the metal uptake has been investigated. The adsorption rates and sorption capacity, desorption ratios, competitive sorption (q cs), values of the partition coefficient (log D) of P(AA/Kry23-DM) SAP were investigated as a function of time with heavy metal ion concentration. The authors were able to show that the crosslinked superabsorber proved to be stable over the entire pH range. The Langmuir isotherm and Freundlich isotherm were in a good agreement with the positive sorption results for the investigated cations. The conformational analysis of the crosslinker and the SAP copolymer was performed using MM2, AMBER and OPLS molecular mechanics.
However, the molecular dynamics results (computer-aided method) were not sufficiently discussed and compared with the literature in the interpretation (see section 4.4) and summary and were not mentioned in the discussion (see section 5.)
  The strength of this work lies in its broad investigation and the consideration of the computer-assisted method, but special attention must be paid to the English language (spelling, style, as well as separate and combined spelling). Furthermore, the quality of the illustrations must be improved in some cases. The interpretation of the data and the added value of computer-assisted methods should be more clearly identified.   The following points need to be improved:   - More recent literature, especially review articles, should be cited (see references 1-4).
- In this context, define and explain the terms "MM2, AMBER and OPLS" in the abstract. Please provide more information about the different methods.
- More literature on the toxicity of metals should be published (see literature reference 15).
- See 2.1: What purity was used for the technical CHCl3?
- See 2.2.1: More details are missing in the recorded FTIR spectra (e.g. which wavenumber range was investigated, scan rate, KBR crystal, etc).
- See 2.2.3: More technical details about the AAS method (manufacturer, measurement conditions, etc) must be given.
- See 2.2.6: More technical details are required for the experimental description of the SEM investigations
- Fig. 1 and 2 must be improved in terms of optical quality. The font size of the axis designations is too small. The shaft numbers (see figure 1) should be placed next to the functional groups.
- Figure 6: The pictures are too small and the size bars are not integrated in the picture
-Figure 9: Were the experiments carried out only once per cation? How large are the deviations? R2 values should be given. The straight lines lie on top of each other and are not easy to see (if necessary, introduce an axis contraction)
- Figure 11: Provide R2 values. Please integrate this into the text and discuss the results.
-Figure 13 should be interpreted more in the text
- Re 3.6: Why were the pH values 3, 7 and 9 selected?
- Re 4.3: Why is "for OH peaks of AA" highlighted (in bold)?
- The molecular dynamics results (computer-based method) were not sufficiently discussed in section 4.4 and compared with the literature and not mentioned in the discussion (see section 5). Why were the computer-aided methods chosen and what conclusions can be drawn? This should be made clearer to the reader.

Round 2

Reviewer 1 Report

Most things have been corrected.

There are still some minor mistakes in the text, e.g.

144: Synthesis of Kry23-DM crosslinker was synthesized

276: p H

549: p H

May be there might be more of these examples, please check carefully.

Scheme 1 has still bad aspect ratio

Please check for uniformity of font size and line pitch

Reviewer 2 Report

Dear authors,

Many thanks for the revised version. Enclosed are further comments that would be important for improving the quality of the manuscript:
- Pay attention to the spelling, font size and sentence structure.
  - Unfortunately the quality of the SEM images (see figure 6) is still very poor. It would be much better for the reader if better pictures were used. Furthermore, figure 6 is mentioned far from the place in the text where the results are discussed.   - The quality of figure 8 needs to be improved (resolution of the figure and size). It would also be better for the reader to explain figure 8 more in detail.

Author Response

Thank you to Reviewer 2 for your corrections and your contribution.

Point 1: - Pay attention to the spelling, font size and sentence structure.  

Response 1:  Spelling, font size and sentence structure were paid attention and corrected.

Point 2: - Unfortunately the quality of the SEM images (see figure 6) is still very poor. It would be much better for the reader if better pictures were used. Furthermore, figure 6 is mentioned far from the place in the text where the results are discussed.  

Response 2:   The quality of SEM images appears better when the image size is enlarged.  I haven’ t got photoshop editing program.

The figures are given in experimental part 3. As the interpretation of the results is given in Section 4 of Conclusion and Discussion, Figure 6 seems far-fetched.

Point 3: - The quality of figure 8 needs to be improved (resolution of the figure and size). It would also be better for the reader to explain figure 8 more in detail.

Response 3: The graph has been redrawn to improve the quality of Figure 8. The resolution and size have been reorganized to make it easier for the reader to understand figure 8. At the same time, Figs 10 and 12 have been revised.